# Long-Term Results of Surgical Treatment and Patient-Reported Outcomes in Congenital Adrenal Hyperplasia—A Multicenter European Registry Study

**DOI:** 10.3390/jcm11154629

**Published:** 2022-08-08

**Authors:** Susanne Krege, Henrik Falhammar, Hildegard Lax, Robert Roehle, Hedi Claahsen-van der Grinten, Barbara Kortmann, Lise Duranteau, Agneta Nordenskjöld

**Affiliations:** 1Department of Urology, Pediatric Urology and Urooncology, Kliniken Essen Mitte, 45136 Essen, Germany; 2Department of Endocrinology, Karolinska University Hospital, 171 77 Stockholm, Sweden; 3Department of Molecular Medicine and Surgery, Karolinska Institutet, 171 76 Stockholm, Sweden; 4Institute of Medical Informatics, Biometry and Epidemiology, University of Essen, 45147 Essen, Germany; 5Institute of Biometry and Clinical Epidemiology, Charite-University Medicine Berlin, 10117 Berlin, Germany; 6Institute of Health, Charite-University Medicine Berlin, 10117 Berlin, Germany; 7Department of Pediatric Endocrinology, Radboud University Medical Center, 6525 GA Nijmegen, The Netherlands; 8Department of Pediatric Urology, Radboud University Medical Center, 6525 GA Nijmegen, The Netherlands; 9Department of Medical Gynaecology and Reference Centre for Rare Diseases of Genital Development, Bicetre Hospital, APHP Paris Saclay University, 94270 Le Kremlin Bicetre, France; 10Department of Women’s and Children’s Health, Center of Molecular Medicine, Karolinska Institutet, 17176 Stockholm, Sweden; 11Department of Pediatric Surgery, Astrid Lindgren Children’s Hospital, Karolinska University Hospital, 17176 Stockholm, Sweden

**Keywords:** 46,XX-DSD, congenital adrenal hyperplasia, surgical interventions, timing of surgery, patient-reported outcomes

## Abstract

Representatives for congenital adrenal hyperplasia (CAH) continue to desire early feminizing surgery in girls with 46,XX-CAH. The aim of this analysis, which included 174 46,XX- individuals with salt-wasting (SW) or simple-virilizing (SV) CAH, a female gender identity, and an age > 16 years participating in a multicenter cross-sectional clinical evaluation study (dsd-LIFE), was to evaluate the long-term results of surgery and patient-reported outcomes (PRO). The gynecological examination (*n* = 84) revealed some shortcomings concerning surgical feminization. A clitoris was absent in 9.5% of cases, while a clitoral hood was missing in 36.7% of cases. Though all women had large labia, they didn’t look normal in 22.6% of cases. Small labia were absent in 23.8% of cases. There was no introitus vaginae, and the urethra and vagina had no separate opening in 5.1% of cases. A mucosal lining was missing in 15.4% of cases. Furthermore, 86.2% of the women had scars at the region of their external genitalia. A vaginal stenosis was described in 16.5% of cases, and a meatal stenosis was described in 2.6% of cases. Additionally, PRO data showed a very-/high satisfaction rate of 21.3%/40.2% with cosmesis and 23.8%/38.1% with functionality, while 3.3%/10.7% showed a very-/low satisfaction with cosmesis as well as 5.6%/10.3% with functionality. The remaining women—24.6% and 23.8%—were indifferent. Satisfaction concerning sex life was very-/high in 9.6%/27.7%. In 12.0%/16.9% it was very-/low. Furthermore, 33.7% had no opinion. Furthermore, 27.0%/31.6% of the women reported that clitoriplasty, but not clitoridectomy, had a very-/positive influence on their lives, while 1.3%/8.9% felt it to be very-/negative, and 28.4% were indifferent. Vaginoplasty had a very-/positive influence in 25.7%/33.8% and a very-/negative effect in 3.6%/6.8%. 29.7% had no opinion. Additionally, 75.7% of the women preferred feminizing surgery during infancy/childhood, especially concerning clitoreduction. In conclusion, though the majority of the participants (76%) preferred early feminizing surgery and 60% described a positive effect on their lives, about 10% felt it to have been negative. About 15% of the women suffered from insufficient cosmesis and functionality after surgery. Sex life was even described as poor in nearly 30%. Therefore, the decision about early genital surgery in 46,XX-CAH girls should be considered carefully. Parents should get detailed information about possible complications of surgery and should receive support to understand that postponing surgery does not inevitably cause harm for their child. Importantly, genital surgery when performed in children should only be performed in expert centers with a specialized team including surgeons who are trained in feminizing surgery.

## 1. Introduction

Congenital adrenal hyperplasia (CAH) is caused by adrenal enzyme defects. Its most common variant, 21-hydroxylase deficiency (21OHD), leads to deficient production of cortisol and, most often, also aldosterone, together with increased steroid precursors and androgens [1]. As a consequence, girls with classic CAH have variably virilized external genitalia at birth, with salt-wasting (SW), if there is a cortisol and aldosterone deficiency, or simply-virilized (SV), where the aldosterone production is not affected. Due to the Chicago classification from 2006, CAH belongs to the differences of sex development (DSD) [2,3]. In most Western countries, early genital surgery with the aim of achieving unambiguous female or male external genitalia in children with DSD was performed for many decades. Virilized girls with 46,XX-CAH have been surgically adapted in female direction. Depending on the severity of virilization, characterized by the Prader classification [4], this included clitoral reduction surgery—in earlier years, clitoris amputation was also performed—as well as the removal of the common urethral and vaginal duct, the so-called urogenital sinus, and the reconstruction of the distal vagina [5].

The parental concerns that atypical genitalia could have an impact on gender identity during infancy might lead parents to endorse early genital surgery. Arguments against surgery are physical damage, caused by poor cosmetic and functional results, and that is partially irreversible, as well as psychological damage in children, who are not able to decide by themselves in favor of surgery [6]. Larger series with regard to these points are hardly available. Therefore, the aim of this European multicenter registry study was to investigate the long-term outcome of genital surgery and the patient-reported outcome concerning satisfaction in women with 46,XX-CAH.

## 2. Methods

### 2.1. Study Design and Participants

This was a multicenter cross-sectional clinical evaluation study (dsd-LIFE), funded by the European Union, to evaluate the clinical care of people with DSD. The study is described in detail elsewhere [7] but is summarized below. The dsd-LIFE consortium consisted of 16 European tertiary centers from Germany, France, the Netherlands, Poland, Sweden, and the United Kingdom (UK), of which 14 were active recruiting sites. Recruitment of persons (≥16 years) with DSD took place during 2014 and 2015. In total, 3100 eligible persons were approached, of whom 1040 agreed to take part in the study. All included people had a DSD condition as described in the classification system of the Chicago Consensus Conference [2,3]. All participants gave written informed consent, and the study was approved by each local ethical committee.

This study had two parts. The first included a retrospective chart review collecting data on age, class of CAH and grade of virilization at diagnosis, and time and type of surgical treatments, as well as a medical interview, with an optional gynecological/urological examination. All tasks were carried out by trained researchers, following standard operation procedures.

In the second part of the study a questionnaire about patient-reported outcomes (PRO) was analyzed.

### 2.2. Outcome Measurements for the Current Study

Standardized genital examination included the evaluation of external genital appearance, such as the size of clitoris or presence/absence of small labia, and functional parameters, such as the depth and width of the vagina. The examination was performed by gynecologists or urologists. Additionally, PRO related to the influence of different kinds of surgery, such as a clitoreduction/clitoridectomy and vaginoplasty, as well as vaginal dilatation, on patients’ lives on a 5-point Likert scale from very negative to very positive. Another group of questions concerned genital and sexual satisfaction. Genital satisfaction was assessed by detailed evaluation of the external genitals and answering the following self-constructed questions: “Are you satisfied with the appearance of your genitals after surgery?”, and “Are you satisfied with how your genitals function after surgery?” With regard to sexual satisfaction, participants were surveyed on their satisfaction with general arousability and orgasmic capacity. Lastly, participants rated their satisfaction with their sex life in general. All questions about satisfaction were rated on a 5-point Likert scale, from very dissatisfied to very satisfied. Regarding the attitudes towards surgery, participants were asked to what extent they agreed that clitoreduction is necessary in girls, if a vaginoplasty in adolescence or adulthood with a patient’s consent is better than before 6 months of age, and what the preferred age for genital surgery in CAH should be (infancy/childhood, or adolescence/adulthood).

### 2.3. Statistical Analyses

Background, surgical data, gynecological examination, and PRO data were displayed as absolute and relative frequencies and mean ± SD or median (range) as appropriate. All proportions were calculated discounting missing values. All analyses were performed using SPSS statistics 22.0.

## 3. Results

In total, 226 persons with 46,XX-CAH (221 with female and 5 with male gender identity) were recruited from several European countries. Of those 221 females, 109 (49.3%) had SW-CAH, 65 (29.4%) had SV-CAH, 33 (14.9%) had NC-CAH, and 14 (6.4%) had a rare form of CAH. Of the five individuals living as males, two had SW-CAH, one SV-CAH, one NC-CAH, and one a rare form of CAH. All of these five persons, except the one with NC-CAH, underwent some kind of genital surgery/Two persons definitively had masculinization surgery, and one was operated early with feminization surgery. The analysis below only considered the 174 girls with SW- and SV-CAH, because in these subgroups feminizing surgery wasmost often performed. At birth, 8/174 (4.6%) girls had clearly female external genitals and 155/174 (89.1%) had ambiguous external genitals. In 11 cases, this information was not mentioned within the charts. Prader classification stages were reported in 112 cases. Based on clinical report forms and patient-reported data, 140/155 girls (93%) underwent operational adjustments in the female direction. Age at first surgery was available in 106 patients. Table 1 shows these basic data for the SW- and SV-CAH girls.

Table 2 shows the distribution of the different surgical procedures per CAH class. The highest percentage of both clitoris surgery and vaginoplasty was seen in the SW-CAH group. Data on the type of vaginoplasty were available for only 73 patients. The most common of these were the use of a Fortunoff flap and pedicled or free skin (*n* = 55, 75.3%). Other procedures performed were labiaplasties (*n* = 53) and perineal plastic surgeries (*n* = 32). A vaginal dilatation was reported in 50 patients, though unfortunately data were missing in the majority of the patients. 

### 3.1. Gynecological Examination

A gynecological examination was performed in 84 women. Median age was 28 years. A clitoris was present in 76 cases, while a clitoral hood was found in only 50 cases of them, and it was hypertrophied in 10 of these cases. Normal looking large labia were present in 65 women, while in another 19 women they rather reminded researchers of a scrotum or looked baggy. Small labia were present in 64 women, while in 20 cases no small labia could be seen. In 14 cases, an asymmetry was described. All details are given in Table 3. 

The overall cosmesis was rated good in 44 cases (54%), satisfactory in 35 cases (43%), and poor in 3 cases (3%) by the physician.

### 3.2. Patient-Reported Outcomes

Women were also surveyed about surgeries. Median age when answering the questionnaire was 28 years (range 15–64). Data differed considerably compared with the data from patients’ charts. While medical charts documented clitoris surgery in 122 persons and vaginoplasty in 121, women themselves reported clitoris surgery in 88 cases, 81 in the form of a clitoris reduction, and 7 as clitoridectomy. Vaginoplasty was reported by 90 females, and 52 women mentioned that they had undergone vaginal dilatation between the age of 9 to 15 years. Information about the specified number of surgical interventions could be given by 114 women The majority had one (40%) or two (34%) surgeries, while 15% underwent three procedures, and 10.5% had more than three operations. However, the information provided does not clearly indicate the nature of subsequent interventions.

Furthermore, 99 out of 174 women (52.9%) reported to have had complications, though in 40 of them no detailed information about the kind of complication was available. Eleven women had more than one complication. Table 4 shows the rate of the most common complications. The highest rate for every complication was reported in SW-CAH. 

Table 5 shows how surgeries and vaginal dilatation influenced the women’s lives. No woman mentioned clitoridectomy to have had a positive influence on her life, while nearly 60% confirmed this for clitoreduction. However, 10% denied this. The same rate of consent, respectively, denial, was reported about vaginoplasty. Concerning vaginal dilatation, 53% of the women said that it had a positive influence on their lives. Only about 11% denied this.

Table 6 summarizes the answers regarding women’s satisfaction with their genitals. A very high or high satisfaction rate was reported concerning cosmesis with 61.5% satisfaction and functionality with 61.9% satisfaction, while 14% and 15.9% of the patients were very unsatisfied or unsatisfied with cosmesis and functionality. Satisfaction concerning sex life was very high or high in 37.3% of the patients, very low or low in 28.9%, and 33.7% of the patients had no opinion.

Additionally, three questions were related to patients’ opinion on the need and timing of feminizing surgery in general. Overall, participants were questioned whether they thought that clitoral reduction surgery is necessary in girls, whether vaginoplasty in adolescence or adulthood with patient`s consent is better than before 6 months of age, and about the appropriate time for genital surgery in general (Table 7). Participants were asked regardless of whether they themselves had had surgery. 

## 4. Discussion

Early genital surgery in DSD has been the subject of debate for many years. Not only do medical outcome parameters play a major role in this, but human rights issues are also a concern. Finally, the discussion has led to many countries, including Germany, to put a general ban on genital surgery for young children with DSD. However, CAH organizations do not agree with such a general ban, because they believe that the gender identity of a girl with CAH generally does not change over time and, therefore, early surgery is justified to prevent harm due to delayed surgery. The majority of persons with 46,XX- CAH identify themselves as female, and continue to prefer early feminizing surgery [8,9,10]. On the other hand there are reports about severely virilized 46,XX-CAH newborns assigned as males during childhood with a stable male identity in adulthood [11]. The aim of this study was to reevaluate these points, specifically surgical outcome, satisfaction rates with cosmesis and function of the genitals, and timing of surgical procedures, within a larger cohort of women with 46,XX-CAH. 

Current surgical genital procedures in CAH include a modification of the clitoris, an introitoplasty and, in cases of a urogenital sinus, lifting of the sinus with reconstruction of the distal vagina. Historical descriptions about clitoral surgery favored clitoridectomy, which remained common and persisted even into the early 1980s [12]. Goodwin described the clitoris reduction with preservation of the glans and neurovascular bundle [13]. This technique underwent further refinements [14,15] and is currently the gold standard. The type of vaginoplasty depends on the Prader stage [4]. While a cut back procedure is sufficient in Prader stage I-II, higher stages require more challenging surgery. A common technique is the Fortunoff flap vaginoplasty combined with the pull-through technique depending on the length of the sinus urogenitalis [16,17]. The development of surgical techniques, especially with the intention to abandon clitoridectomy, suggest a better outcome of surgery nowadays. However, small series reporting long-term results about cosmesis and functionality after feminization do not really reflect this. Creighton et al. published long-term results of 44 adult women who underwent surgery in the early 1980s, of which 41% had a poor cosmetic result. Reintervention, mostly because of introitus or intravaginal stenosis, was necessary in 23 of 26 (89%) patients who received vaginoplasty [18]. Insufficient functionality was described by Crouch et al. and Minto et al. All patients with clitoral surgery had decreased thermal and vibratory sensitivity compared to patients without surgery and controls. All participants also received a sexual function questionnaire, which showed that patients with surgery had a lower frequency of intercourse, vaginal penetration difficulties, and a higher rate of anorgasmia [19,20]. Sircili et al. presented long-term data from 34 patients, who underwent single-stage clitorovaginoplasty between 1986 and 2002. The neurovascular bundle was preserved in only seven cases. In the other 27 patients, blood supply was only maintained by the ventral mucosa. At examination, a clitoris was visible in all 7 patients with preservation of the neurovascular bundle, but only in 21 of the other patients. Persistence of the urogenital sinus was found in 11 patients. Functional results considering menstrual flow, appearance of the introitus, and possibility of sexual intercourse were judged as excellent in 67%, good in 25%, and regular in 8% of patients [21]. Van der Zwan et al. reported about 40 patients. Of these, 36 underwent feminizing surgery, 13 as a single-stage clitorovaginoplasty at a median age of 3 years. Seven of them (54%) needed resurgery. Twenty patients underwent a two-stage procedure, clitoriplasty in early childhood, and vaginoplasty at a median age of 13. In this group, several patients needed additional surgery. In summary, 25 of the 36 patients (69%) underwent redo-operations [22]. 

Concerning the time for surgery, the majority of physicians preferred one-stage surgery in early childhood. Arguments from the medical side are that the tissue still possesses high elasticity in the first 3–6 months of life due to the continuing influence of the maternal estrogens, as well as the use of excess skin in clitoral reduction surgery for vaginal plastic surgery [23,24,25,26,27,28]. From a psychological point of view, the clarity of the child’s genitals provides relief for parents [29]. A smaller group of surgeons favored a two-stage procedure, whereby only clitoris reduction is carried out in early childhood and a vaginoplasty is performed only at the beginning of puberty. Reasons for this are lack of indication of a vaginal reconstruction before menarche, risk of vaginal constriction caused by inadequate stretching/dilatation after early vaginal reconstruction, and long-term risk of vaginal stenosis [30,31,32,33]. If the vaginoplasty takes place only at the beginning of puberty, the girls can be trained in dilatation performed by themselves. Finally, it should be mentioned that the decision of clitoreduction within the first 6 months of life does not take into account the often remarkable effect of clitoris reduction during medical treatment. 

In our study, 140/155 (93%) of the females with ambiguous genitalia underwent genital surgery. Most commonly, the time for surgery was the first year of life (57/106; 53%) or within the first 4 years (34/106; 32%). In total, 71/119 (60%) underwent a one-stage procedure, while 48/119 (40%) had a two-stage procedure. In our study, 8.5% of the girls with clitoral surgery underwent a clitoridectomy. Although the gynecological examination revealed some shortcomings, the evaluation of the surgical results by the physicians was positive in 97% of cases. However, 14% of the women were (very) unsatisfied with the cosmesis (17/122) and functionality (18/126). Sex life in general was described as (very) satisfying by only 37% (62/166) of the women. This shows that there are discrepancies between physicians and affected persons themselves when it comes to judging the surgical results, and it underlines the necessity of patient-reported outcomes, because their subjective opinion is the most important. However, the satisfaction rates may also highlight an earlier lack of information about the physical condition at birth, since females compare themselves with healthy women.

In general, surgery and even vaginal dilatation had a positive influence in about 60% of the women, except in those with clitoridectomy, which was evaluated negatively in 86% of cases. Vaginal dilatation was performed by the young women themselves between the ages of 9 to 15 years. At this time, they can understand the advantage of dilatation and will be interested in having a functioning vagina. Concerning the right time for surgery, 76% (109/144) voted for infancy and childhood, while only 10% (14/144) preferred adolescence or adulthood. When especially asked if clitoreduction is necessary in girls, 69% (92/133) agreed, but only 35% (43/122) also voted for vaginoplasty during the first 6 months of life, while 49% (60/122) preferred the latter procedure with the patient’s consent.

In conclusion, this study underlines that the surgical outcome of feminizing procedures in individuals with 46,XX CAH can be optimized. Therefore, only surgeons trained in this kind of surgery should perform these procedures. Prospective registration of surgical procedures and their outcome may improve knowledge. Counseling about feminizing surgery in individuals with 46,XX-CAH should be performed in expert centers with -an multidisciplinary team to provide informed consent, especially in cases of a decision for early surgery. 

The study has drawbacks since it has a cross-sectional design, and data retrieved retrospectively from the medical files are incomplete. Moreover, there is an incongruence among those data from the medical files and what patients remember, which can be seen when comparing the number of clitoral surgeries and vaginoplasties documented in the charts and what patients report. However, this might reflect the former policy not to involve patients in their situation so that they were not informed about early surgeries they underwent and do not remember. An important advantage of the study are the data of the gynecological examination and the patient-reported outcomes, though a possible bias might be that patients were primarily recruited via clinics that seem to have good experience with surgery in DSD. The gynecological examination enables a comparison about how the physicians rated the surgical results on the one side and the patients on the other side. Finally, the patients themselves gave statements about their opinions concerning time and the kind of surgery necessary. Moreover, the absence of the evaluation between specific surgical procedures and specific centers with the gynecological outcome and PRO is a great limitation.

## Figures and Tables

**Table 1 jcm-11-04629-t001:** Basic data of 174 girls with 46,XX-CAH, subdivided into SW- and SV-CAH.

**1a: Prevalence of Prader grade by CAH class**					
	**Class**	**SW**		**SV**	
	** *n* **	** *n* **	**(%)**	** *n* **	**(%)**
**Prader grade**	**174**	**109**	**(62.6)**	**65**	**(37.4)**
I	3	1	(33.3)	2	(66.7)
II	17	5	(29.4)	12	(70.6)
III	33	18	(54.5)	15	(45.5)
IV	50	42	(84.0)	8	(16.0)
V	9	9	(100.0)	0	
Unknown	62	34	(54.8)	28	(45.2)
**1b: Proportion of genital surgery per CAH class**					
	**Class**	**SW**		**SV**	
	** *n* **	** *n* **	**(%)**	** *n* **	**(%)**
**Surgery**	**174**	**109**	**(62.6)**	**65**	**(37.4)**
Yes	140	89	(63.6)	51	(36.4)
**1c: Distribution of one-/two-stage genital surgery per CAH class**					
	**Class**	**SW**		**SV**	
** *n* **		** *n* **	**(%)**	** *n* **	**(%)**
**I/II-stageOP**	**174**	**109**	**(62.6)**	**65**	**(37.4)**
One stage	71	42	(59.2)	29	(40.8)
Two stage	48	35	(72.9)	13	(27.1)
Unknown	55	32	(58.2)	23	(41.8)
**1d: Age at first genital surgery per CAH class**					
	**Class**	**SW**		**SV**	
	** *n* **	** *n* **	**(%)**	** *n* **	**(%)**
**Age at 1. OP**	**174**	**109**	**(62.6)**	**65**	**(37.4)**
</= 1 year	57	41	(71.9)	16	(28.1)
2–5 years	34	21	(61.8)	13	(38.2)
6–12 years	7	2	(28.6)	5	(71.4)
13–29 years	8	6	(75.0)	2	(25.0)
Unknown	68	39	(57.3)	29	(42.7)

**Table 2 jcm-11-04629-t002:** Proportion of surgeries per CAH class.

	Class	SW		SV	
	*n*	*n*	(%)	*n*	(%)
Surgery		109	(62.6)	65	(37.4)
Clitoridectomy	10	6	(60.0)	4	(40.0)
Clitoreduction	112	80	(71.4)	32	(28.6)
Vaginoplasty	121	86	(71.1)	35	(28.9)

**Table 3 jcm-11-04629-t003:** Results of the gynecological examination in 84 women with 46,XX-CAH.

Parameter	N	Yes	(%)	No	(%)
Clitoris	84	76	(90.5)	8	(9.5)
Clitoris hood	79	50	(63.3)	29	(36.7)
Large labia	84	84	(100.0)		
Small labia	84	64	(76.2)	20	(23.8)
Introitus vaginae	78	74	(94.9)	4	(5.1)
Mucosal lining	78	66	(84.6)	12	(15.4)
Separate opening for urethra and vagina	78	74	(94.9)	4	(5.1)
Vaginal length *	64	64			
Vaginal width **	73	73			
Vaginal stenosis	79	13	(16.5)	66	(83.5)
Urethrovaginal fistula	77			77	(100.0)
Meatal stenosis	77	2	(2.6)	75	(97.4)
Scars ***	80	69	(86.2)	11	(13.8)

*n* = number of evaluable patients. * <8 cm, *n* = 30 (47%); 8–10 cm, *n* = 31 (48%); 10–12 cm, *n* = 3 (5%). ** <1 finger, *n* = 3 (4%); one finger, *n* = 21 (29%); two fingers, *n* = 49 (67%). *** minimal, *n* = 45 (65.5%); moderate, *n* = 23 (33%); severe *n* = 1 (1.5%).

**Table 4 jcm-11-04629-t004:** Overview of the most common complications according to CAH-classes.

	Class	SW		SV	
	*n*	*n*	(%)	*n*	(%)
Complication		109	(62.6)	65	(37.4)
Vaginal stricture	21	14	(66.7)	7	(33.3)
Scars	18	13	(72.2)	5	(27.8)
Urogenital infection	13	8	(61.5)	5	(38.5)
Urine dribbling	7	6	(85.7)	1	(14.3)

*n* = number of reported complications.

**Table 5 jcm-11-04629-t005:** Influence of surgeries and vaginal dilatation on the lives of women with 46,XX-CAH.

		Very Neg		Neg		Indifferent		Pos		Very Pos	
Procedure	*n*	*n*	(%)	*n*	(%)	*n*	(%)	*n*	(%)	*n*	(%)
Clitorireduction	74	1	(1.3)	7	(8.9)	21	(28.4)	25	(31.6)	20	(27.0)
Clitoridectomy	7	3	(42.9)	3	(42.9)	1	(14.3)	0		0	
Vaginoplasty	74	3	(3.6)	5	(6.8)	22	(29.7)	25	(33.8)	19	(25.7)
Vaginal dilatation	47	0		5	(10.6)	17	(33.3)	13	(27.7)	12	(25.5)

*n* = number of evaluable patients.

**Table 6 jcm-11-04629-t006:** Satisfaction with their genitals in women with 46,XX-CAH.

		Very Low		Low		Indifferent		High		Very High	
Parameter	*n*	*n*	(%)	*n*	(%)	*n*	(%)	*n*	(%)	*n*	(%)
Clitoris size	137	11	(8.0)	9	(6.6)	46	(33.6)	55	(40.1)	16	(11.7)
Clitoris shape	134	13	(9.7)	3	(2.2)	43	(32.1)	57	(42.5)	18	(13.4)
Sexual arousal	140	4	(2.9)	15	(10.7)	33	(23.6)	62	(44.3)	26	(18.6)
Orgasm	139	16	(11.5)	19	(13.7)	34	(24.5)	54	(38.8)	16	(11.5)
Vaginal length	140	0		0		46	(32.9)	73	(52.1)	21	(15.0)
Vaginal width	142	7	(4.9)	9	(6.3)	42	(29.6)	67	(47.2)	17	(12.0)
Vaginal moisture	142	5	(3.5)	28	(19.7)	40	(28.2)	58	(40.8)	11	(7.7)
Functionality	126	7	(5.6)	11	(10.3)	30	(23.8)	48	(38.1)	30	(23.8)
Cosmesis	122	4	(3.3)	13	(10.7)	30	(24.6)	49	(40.2)	26	(21.3)
Sex life	166	20	(12.0)	28	(16.9)	56	(33.7)	46	(27.7)	16	(9.6)

*n* = number of evaluable patients.

**Table 7 jcm-11-04629-t007:** Questions concerning need and timing of feminizing surgery in women with 46,XX-CAH.

		Agree ++		Agree +		Indifferent		Disagree −		Disagree −	
Question	*n*	*n*	(%)	*n*	(%)	*n*	(%)	*n*	(%)	*n*	(%)
Need for clitoral reduction in girls	133	64	(48.1)	28	(21.0)	30	(22.6)	4	(3.0)	7	(5.3)
Vaginoplasty with patients’ consent better than before 6 months	122	25	(20.5)	18	(14.7)	19	(15.6)	22	(18.0)	38	(31.1)
		Infancy		Childhood		Indifferent		Adolescence		Adulthood	
		*n*	(%)	*n*	(%)	*n*	(%)	*n*	(%)	*n*	(%)
Appropriate time for genital surgery	144	76	(52.8)	33	(22.9)	21	(14.6)	8	(5.6)	6	(4.2)

*n* = number of evaluable patients.

## Data Availability

The datasets generated and analysed during the current study are not publicly available because data analysis will mainly be performed by the dsd-LIFE consortium. The data presented in this study are available on reasonable request from the corresponding author.

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
