# Peer review of "Long-Term Results of Surgical Treatment and Patient-Reported Outcomes in Congenital Adrenal Hyperplasia—A Multicenter European Registry Study"

_jcm, 2022, doi:10.3390/jcm11154629_

Round 1

Reviewer 1 Report

This is a very important study in an area where data from large cohorts is lacking. The authors present data on female-identified individuals with various types of CAH from the dsd-LIFE study and evaluate outcomes and attitudes towards surgery, including patient-reported outcomes. 

- I do think it would be valuable to at least have one paragraph summarizing the results/attitudes/well-being of the 5 people with a male gender identity. I don't think they need to be in the tables, but so little is known about this group, it would be nice to have a short paragraph. 

- Table 1b does not show correlations as the title suggests, it just shows prevalence of prader grades by CAH type. Re-word the title 

- Give more detail as to how patients were classified as having SW, SV or NC CAH 

- What was the median age of assessment for the outcomes in Table 4? 

- How many people had a repeat surgery for a complication? 

- On page 6, it would be nice to know how many people only had 1 complication vs >1 complication 

- What percent of people were not sexually active at all? 

- The discussion gives lots of detail about specific studies and could be strengthened with an overall narrative for each section 

Minor comments 

- Recommending using people-first language, for example "People with CAH" rather than "CAH patients." At some points, the authors use people-first language, at others they do not 

- I'd use "sex" instead of "sexual" - "differences of sex development" instead of "differences of sexual development" 

- In tables 1c-e specify that this is "genital" surgery

- In table 3, do we think the person who had a vaginoplasty with non-classic CAH really had non-classic CAH

- I am less familiar with the term "bouginage" (page 8) - is this dilation? 

Author Response

Cover letter concerning reviewers‘ comments to manuscript jcm-1813380

Dear ladies and gentlemen,

first of all we want to thank you for the revision of our manuscript.

We have integrated the comments of the two reviewers in the manuscript, which is described in detail here and marked in the text. As reviewer 2 strongly recommended only to consider the girls with salt-wasting and simple virilizing CAH, because the paper focuses on surgery, which is mainly performed in these two subgroups, we have done a completely new analysis of all parameters for these two groups exclusively. We made new tables instead of marking the changes, because this would be too confusing.

Reviewer 2 also annotated that there are too many tables. Therefore table 1a and table 2 have been removed and the information is given in the text.

There is one new reference and some after shortening of the discussion have been deleted. The new order of the literature has also been done without marking the changes for better reading.

Reviewer 1:

Major comments:

1 Recommendation to have a short paragraph about the five 46,XX persons with CAH and male identity

Some information about these 5 persons is added in the first paragraph of the Results.

2 Recommendation to re-word the title of table 1b

The title of table 1b has now been rephrased as recommended.

3 Recommendation to give more details how patients were classified as having SW, SV or NC CAH

Within the Introduction we describe the differences of the subgroups of CAH. Otherwise the girls were classified as it was documented within the medical charts.

4 What was the median age of assessment for the outcomes in Table 4?

The median age was 28 years.

5 How many people had a repeat surgery for a complication?

We agree that it would be very interesting to know how many people had a repeat surgery for a complication. Unfortunately, we do not have the information available due to the retrospective nature of the study. The patients’ medical/surgical charts mostly included only surgeries during childhood. The necessity for repeat surgeries due to issues like vaginal stenosis mostly becomes obvious after puberty.

In the PRO questionnaire people were only asked if they remember “how many surgical procedures related to their condition they had”. Another question was “ if they had complications”. Therefore it is not possible to correlate repeat surgeries and complications.

The first paragraph under Patient reported outcomes deals with this problem.

6 How many people had 1 complication vs > 1 complication?

This information is added now on page 8.

7 What percent of people were not sexually active at all?

Unfortunately there was no question especially asking about sexual activity at all.

8 Recommendation to strengthen the discussion.  

The discussion has been strengthened due to the recommendation.

Minor comments:

1 Recommendation using people-first language

The text was rephrased, instead of “patients” we speak about “girls or women” with CAH depending on the context.

Nevertheless we think we could use the term patient talking about CAH, because these persons accept that they have a disease, an enzyme defect within the adrenals.

2 Recommendation using “sex” instead of “sexual”

“Sexual” has been changed to “sex” within the manuscript.

3 Considering the person with NC-CAH and vaginoplasty, had this person really a NC-CAH?

This point is no more important, because the revised manuscript only considers the subgroups of SW- and SV-CAH.

4 Considering the term “bouginage”

“Bouginage” means dilatation. The term has been changed to vaginal Dilatation.

Reviewer 2 Report

05.07.2022

To Ms. Ivana Radanovic
Assistant Editor, MDPI Novi Sad,

RE: Manuscript ID: jcm-1813380

In their study entitled "Long-term results of surgical treatment and patient-reported outcomes
in congenital adrenal hyperplasia - a multicenter European registry study" the authors
describe the long-term results of genital feminizing surgery and patient reported outcomes, of rather large group of 46,XX-CAH adult female. In view of the on-going dispute on the ideal age/method of genital surgery in XX females with classic CAH, the current survey is of great value.

The survey shows that the majority of the participants reported high satisfaction rate with both cosmetics and functionality. However, significant minority reported dissatisfaction with sexual life and surgery results, emphasizing the need for improving surgical techniques and performing these complicated procedures in specialized centers.

Comments:

The paper is long and need to undergo thorough editing. Sometime the literature review is more elaborated than needed and unrelated to the current study.

Abstract:

1.    "The gynaecological examination revealed some shortcomings concerning surgical feminization". Description of the shortcomings is lacking.

2.    "Data showed a very-/high satisfaction rate of about 60%", the very high is probably inappropriate

3.     "Parents should get detailed information about the risks of early surgery". As the results were not compared between different ages of intervention, the ward "risks: might be inappropriate

4.    Last line: Many CAH advocacy groups don't want to be included under the DSD umbrella.  Therefore, specialized CAH centers instead of DSD centers might be more appropriate

Introduction:

1.    The description of the different types of DSD is unnecessary. CAH is a distinct category.

2.    46XX-DSD might be replaced by 46XX-CAH.

3.    "As a consequence 46,XX-DSD children with classic CAH". Surgery is only relevant to girls with CAH, therefore girls or females patients with CAH is more appropriate than children.  

4.    First paragraph, line 8 from bottom: " non-classical (NC) CAH is a milder form with only very mild virilization if any". I suggest adding "postnatal" before "virilization"

5.    "Other rare variants of CAH exist with varied virilization (4)". This is not related to the current study and may be omitted.

6.    "In most western countries the external genitalia of virilized children". Children should be replaced by girls.

7.    "Depending on the severity of virilization, characterized by the Prader classification". Refernce describing the Prader classification should be added.

8.    "In CAH patients raised as females who underwent early surgery, in most cases feminizing procedures, gender identity dysphoria is 4% (7-9)". This statement might suggest that age of genital surgery influence future gender identity, which is not the case. It might be replaced by "large series of gender identity in XX CAH females report on gender dysphoria in only 4%.

Methods:

1.    There are many repeats, should be edited.

Results:

1.      "Of the included patients, 109 (49,3%) had SW 21OHD, 65 (29,4%) SV 21OHD, 33 (14,9%) NC 21OHD, 3 (1,4%) unknown 21OHD phenotype……". Since this survey aimes to investigate the outcome of early surgery in virilized girls with CAH, subjects with the non-classic form or rare forms of CAH should be excluded

2.      Table 1d-1e, Table 3. Why NCCAH were operated on? The definition of the non-classic form is post-natal virilization. I strongly suggest that the non-classic subjects will be excluded from the study and the tables

3.      There are too many tables, the most informative should be chosen.

Discussion

1.      "Surgical procedures performed in the past were often cosmetically inadequate and led to functional limitations". Reference is missing

2.      "Feminizing surgery in CAH", first paragraph: the long description of the different surgical methods is unrelated to the current study since correlation between the specific procedures and outcome was not performed

3.      Drawbacks (last paragraph): the absence of evaluation of the association between  specific surgical procedures and specific centers with PRO and gynecologic outcome is a great limitation and should be added.

Author Response

Cover letter concerning reviewers‘ comments to manuscript jcm-1813380

Dear ladies and gentlemen,

first of all we want to thank you for the revision of our manuscript.

We have integrated the comments of the two reviewers in the manuscript, which is described in detail here and marked in the text. As reviewer 2 strongly recommended only to consider the girls with salt-wasting and simple virilizing CAH, because the paper focuses on surgery, which is mainly performed in these two subgroups, we have done a completely new analysis of all parameters for these two groups exclusively. We made new tables instead of marking the changes, because this would be too confusing.

Reviewer 2 also annotated that there are too many tables. Therefore table 1a and table 2 have been removed and the information is given in the text.

There is one new reference and some after shortening of the discussion have been deleted. The new order of the literature has also been done without marking the changes for better reading.

Reviewer 2:

Abstract:

1 Recommendation to describe the shortcomings at the gynecological examination.

The shortcomings described at the gynecological examination have now been added to the abstract as suggested.

2 Comment about the formulation “very-/high satisfaction”

The satisfaction rates from the PROs are given in detail now.

3 Comment about the sentence “Parents should get detailed information about the risks of early surgery” and the missing comparison between different ages of intervention

It is right that surgical results between different ages were not compared, therefore the sentence “Parents should get detailed information about the risks of early surgery” was modified in “Parents should get detailed information about possible complications of surgery”.

4 Recommendation to change the last sentence to “Many CAH advocacy groups don`t want to be included under the DSD umbrella. Therefore, specialized CAH centers instead of DSD centers might be more appropriate.”.

According to the Chicago classification CAH belongs to the variants of DSD. Therefore we have modified the last sentence to “…genital surgery, when performed in children, should only be done in expert centers with a specialized team including surgeons who are trained in feminizing surgery”.

Introduction:

1 Recommendation to remove the paragraph about the different types of DSD

It is right that the paper only present data of one group of DSD people, namely CAH. But especially because of the discrepancy among advocacy groups for CAH and 46,XY-DSD variants voting for or against early surgery, we found that it is necessary to give a short introduction about DSD.

2 and 3 Recommendation to change “46,XX-DSD” by “46,XX-CAH” as well as changing “46,XX-DSD children” by “girls or females with CAH”

The wording has been changed as suggested, i.e. instead of “46,XX-DSD children with classic CAH” the new version has “girls or women with classic CAH”

4 Recommendation to modify the sentence about non-classical CAH

The sentence about non-classical CAH was changed completely, because also reviewer 1 wanted a better definition.

5 Recommendation to omit the sentence “Other rare variants of CAH exist with varied virilization.“

The sentence about rare forms of CAH has been deleted.

6 Recommendation not to speak about “virilized children” but “virilized girls”

“Children” has been replaced by “girls”.

7 Recommendation to add a reference about the Prader classification

A reference about the Prader-classification has been added.

8 Replacement of the sentence about gender identity dysphoria “In CAH patients raised as females …… gender identity dysphoria is 4%.” By “Large series of gender identity in XX CAH females report on gender dysphoria in only 4%.”

The suggested correction has been performed.

Methods:

1 Recommendation to edit repeats

This paragraph was shortened, deleting repeats.

Results:

1 Recommendation to concentrate on the CAH-subgroups SW- and SV-CAH

Patients with non-classical CAH or a rare form were excluded. All tables are new considering only SW- and SV-CAH.

2 Recommendation to reduce the number of tables

According to this recommendation table 1a and table 2 have been removed and the information is given within the text.

Discussion:

1 Hint to add references concerning cosmetically and functionally inadequate surgical results

References about insufficient outcome have now been added. This was not done before, because the first paragraph of the discussion is more general, then the different topics (surgical methods, outcome) are described in detail with references. But we have added the references now already in the first paragraph about surgical outcome.

2 Recommendation to shorten the part “Feminizing surgery in CAH”

The paragraph about the different surgical methods has been shortened.

3 Recommendation to add a limitation about the absence of evaluation of the association between specific surgical procedures and specific centers with PRO and gynecologic outcome

We have now added to the limitation as suggested “Moreover, the absence of the evaluation between specific surgical procedures and specific centers with PRO and gynecologic outcome is a great limitation.

Round 2

Reviewer 2 Report

 Try to further shorten the paper. Any discussion on other DSD diagnosis is irrelevant. Focus on CAH only. 

Author Response

Dear collegue,

we have further shortened the manuscript deleting all not about CAH. There is a small change at the beginning of the abstract (see comment). The introduction and the discussion have been shortened to a larger amount. To make it readable and for comparision with the first revision we have written a new introduction and discussion with the changes of the first revision accepted and added the new parts after the primary introduction and the primary discussion.

Order of references have already been adapted within the text.

Thanks for your support optimizing the manuscript.